# Identification of small-molecule ion channel modulators in *C. elegans* channelopathy models

Qiang Jiang[1,2], Kai Li[1,2], Wen-Jing Lu[3], Shuang Li[2,4], Xin Chen[1,2,5], Xi-Juan Liu[1], Jie Yuan[1,2], Qiurong Ding ![ORCID] [4], Feng Lan[3] & Shi-Qing Cai[1]

Ion channels are important therapeutic targets, but the discovery of ion channel drugs remains challenging due to a lack of assays that allow high-throughput screening in the physiological context. Here we report *C. elegans* phenotype-based methods for screening ion channel drugs. Expression of modified human *ether-a-go-go-related* gene (hERG) potassium channels in *C. elegans* results in egg-laying and locomotive defects, which offer indicators for screening small-molecule channel modulators. Screening in worms expressing hERG[A561V], which carries a trafficking-defective mutation A561V known to associate with long-QT syndrome, identifies two functional correctors Prostratin and ingenol-3,20-dibenzoate. These compounds activate PKCε signaling and consequently phosphorylate S606 at the pore region of the channel to promote hERG[A561V] trafficking to the plasma membrane. Importantly, the compounds correct electrophysiological abnormalities in hiPSC-derived cardiomyocytes bearing a heterozygous CRISPR/Cas9-edited hERG[A561V]. Thus, we have developed an in vivo high-throughput method for screening compounds that have therapeutic potential in treating channelopathies.

[1] Institute of Neuroscience and State Key Laboratory of Neuroscience, CAS Center for Excellence in Brain Science and Intelligence Technology, Shanghai Institutes for Biological Sciences, Chinese Academy of Sciences, 200031 Shanghai, China. [2] University of Chinese Academy of Sciences, 100049 Beijing, China. [3] Beijing Laboratory for Cardiovascular Precision Medicine, Beijing Anzhen Hospital, Capital Medical University, 100029 Beijing, China. [4] CAS Key Laboratory of Nutrition, Metabolism and Food Safety, Shanghai Institute of Nutrition and Health, Shanghai Institutes for Biological Sciences, Chinese Academy of Sciences, Shanghai 200031, China. [5] Present address: Developmental and Stem Cell Program, Arthur and Sonia Labatt Brain Tumor Research Centre, Hospital for Sick Children, Toronto M5G 1X8 ON, Canada. These authors contributed equally: Qiang Jiang, Kai Li. Correspondence and requests for materials should be addressed to F.L. (email: fenglan@ccmu.edu.cn) or to S.-Q.C. (email: sqcai@ion.ac.cn)

on channels are the molecular basis for the cell excitability. Malfunction of ion channels causes over 55 different inherited human diseases known as channelopathies[1]. Although 15% of US Food and Drug Administration (FDA)-approved drugs act through targeting ion channels[2,3], there remains an urgent need for developing drugs for many untreated channelopathies. Discovery of new drugs targeting ion channels is known to be notoriously difficult. Assaying ion channel activity with patch-clamp recording of ion channel currents has very low screening throughput, although this problem is now largely alleviated by recent advances in cell line-based fluorescent assays and automated electrophysiology[2,3]. However, existing in vitro assays using cell lines expressing a particular ion channel could not overcome the problem that the channel in their native tissues exists in a cellular and physiological environment substantially different from that in the cell line, hence exhibits different molecular and functional properties. For example, accessory subunits usually dramatically alter the pharmacological properties of ion channels[4,5]. An efficient screening tool thus should examine the function of ion channels in the physiological context.

Most types of ion channels are found in *C. elegans* and are known to regulate worm behaviors[6–10]. Like their human homologs, *C. elegans* channels usually consist of pore-forming and accessory subunits[10,11]. Many mechanisms underlying the biogenesis of ion channels are conserved from *C. elegans* to humans[11–13]. Based on these findings and cumulative knowledge regarding ion channel mutations associated with human diseases, we hypothesized that expression of disease-relevant human ion channels in *C. elegans* would disturb normal electrical signaling and hence cause phenotypic defects in the transgenic worms. Compounds that modulate human ion channel function could be identified by examining their effects on the phenotypes of the transgenic worms. Phenotype-based screening in *C. elegans* has been used in dissecting mechanism of drug action and screening for new drugs[14–16]. In this study, we specifically targeted ion channel function using phenotype-based screening in *C. elegans* models of channelopathies, and demonstrated that specific small-molecule modulators could be identified for correcting the defective function of mutant ion channels associated with channelopathies, e.g., Long-QT syndrome (LQTS).

Human *ether-a-go-go-related* gene[17] (hERG) encodes a voltage-gated $K^+$ channel that mediates cardiac $I_{Kr}$ currents. Dysfunction of hERG $K^+$ channels due to genetic mutations or drug-induced inhibition results in LQTS[18–22], a life-threatening cardiac arrhythmia that affects 1 in 2500 live births and results in about 5000 death each year in US alone[23]. Due to the fatal side effects of causing LQTS, several drugs have been withdrawn from the market since 1990s and the FDA now requires to test the drug effect on hERG activity during preclinical safety evaluation[22,24]. Most of hERG mutations underlying type 2 LQTS (LQTS2) suppress $I_{Kr}$ currents by causing trafficking defects[25,26]. The mutant hERG $K^+$ channels are usually able to elicit normal or near normal $I_{Kr}$ currents if they are properly transported to the plasma membrane by experimental manipulations, e.g., lowering the incubation temperature[27,28]. Pharmacological correction of the trafficking defect thus represents a new promising strategy for treating LQTS.

In this study, we generate channelopathy animal models by expressing chimeric hERG or trafficking-defective hERG mutant channels in *C. elegans*. Phenotype-based screens in the transgenic worms identify alphitolic acid (ALA) as a novel hERG trafficking inhibitor and two protein kinase C (PKC) activators, Prostratin and ingenol-3,20-dibenzoate (IDB), as functional correctors of some LQTS2 mutant channels. Thus, we provide an in vivo system for the discovery of ion channel modulators.

## Results

**Generation of *C. elegans* animal models of channelopathies.** Human ion channels share high similarity with their *C. elegans* homologs, especially in their transmembrane domains[29]. To overcome the possible barrier of expressing human ion channels in *C. elegans*, we first made human-worm chimeric ion channels according to the strategy described in Fig. 1a. Ion channel inhibitors and activators often target at transmembrane domains and/or loops at the membrane periphery of the channels. We thus assumed that a chimeric ion channel consisting of transmembrane domains and their connecting loops of a human ion channel, and N-/C-terminus of its *C. elegans* homolog, would largely maintain pharmacological properties of the human ion channel and could be functionally expressed in *C. elegans*. A gain-of-function mutation was introduced into the channel (named as chimeric ion channel 1, Fig. 1a) to induce clear behavioral defects in the transgenic worms expressing the channel, and drugs that alleviate the phenotypic defects could be inhibitors of the channel (Fig. 1b). For screening channel activators or functional correctors of a mutant channel, a second mutation, which is a loss-of-function mutation, was introduced into the channel (named as chimeric ion channel 2, Fig. 1a), and transgenic worms expressing this channel should show no or weak behavioral defects due to the reduced channel function. Small molecules that aggravate the worms' behavioral defects are potential activators of the channel (Fig. 1b).

According to the above strategy, we made *C. elegans* models of channelopathies for identifying chemical modulators of hERG $K^+$ channels. The hERG $K^+$ channel shows homology with the *C. elegans* ERG-type $K^+$ channel UNC-103 (Supplementary Fig. 1a), which is widely expressed in *C. elegans* head neurons and vulva muscles and regulates worms' locomotion, egg-laying, and male mating behaviors[6,7]. We first tried to express full-length hERG $K^+$ channels in *C. elegans*, but failed due to mistrafficking of the channel. We then constructed a GFP fused chimeric hERG (hERG^chimera/A536W::GFP), which consisted of N-terminus and C-terminus of UNC-103, transmembrane, and cyclic nucleotide-binding domains of hERG, and a gain-of-function mutation A536W[30] in the S4 segment of the hERG $K^+$ channel (Supplementary Fig. 1b). The mutation A536W, which switches the voltage-dependent activation of hERG $K^+$ channels to a much more negative potential[30], was expected to alter cell excitability and consequently induce behavioral defects in worms. Indeed, expression of *unc-103* promoter-driven hERG^chimera/A536W::GFP proteins in N2 wild-type worms generated transgenic worms (hERG^chimera/A536W) with clear defects in locomotion and egg-laying (Fig. 1c–e). Fluorescent images suggested that hERG^chimera/A536W::GFP channels could be delivered to the plasma membrane (Fig. 1f). In addition, expression of hERG^chimera/A536W rescued the copulatory spicule protraction defect caused by a loss-of-function mutation in the *unc-103* gene in male worms (Supplementary Fig. 1c). Taken together, these data suggest that hERG^chimera/A536W could affect worms' behaviors.

Next, we examined whether behavioral defects in worm models of channelopathies could be rescued by known specific blockers of these channels. Cells of *C. elegans* are relatively inaccessible for most chemical compounds because of their cuticles and xenobiotic efflux pumps in the intestine. To facilitate penetration of compounds into the worm, we generated *acs-20;hERG^chimera/A536W* mutant worms by crossing *hERG^chimera/A536W* with *acs-20* null mutant worms, whose cuticles have increased permeability for small molecules[31]. To further enhance the targeting of the compounds, we fed these worms with bacteria expressing double-strand RNAs targeting *ifd-2* and *c15c7.5*, two genes encoding nonessential intestinal intermediate filament proteins that impede the effects of ingested compounds[32]. We found that the phenotypes in the

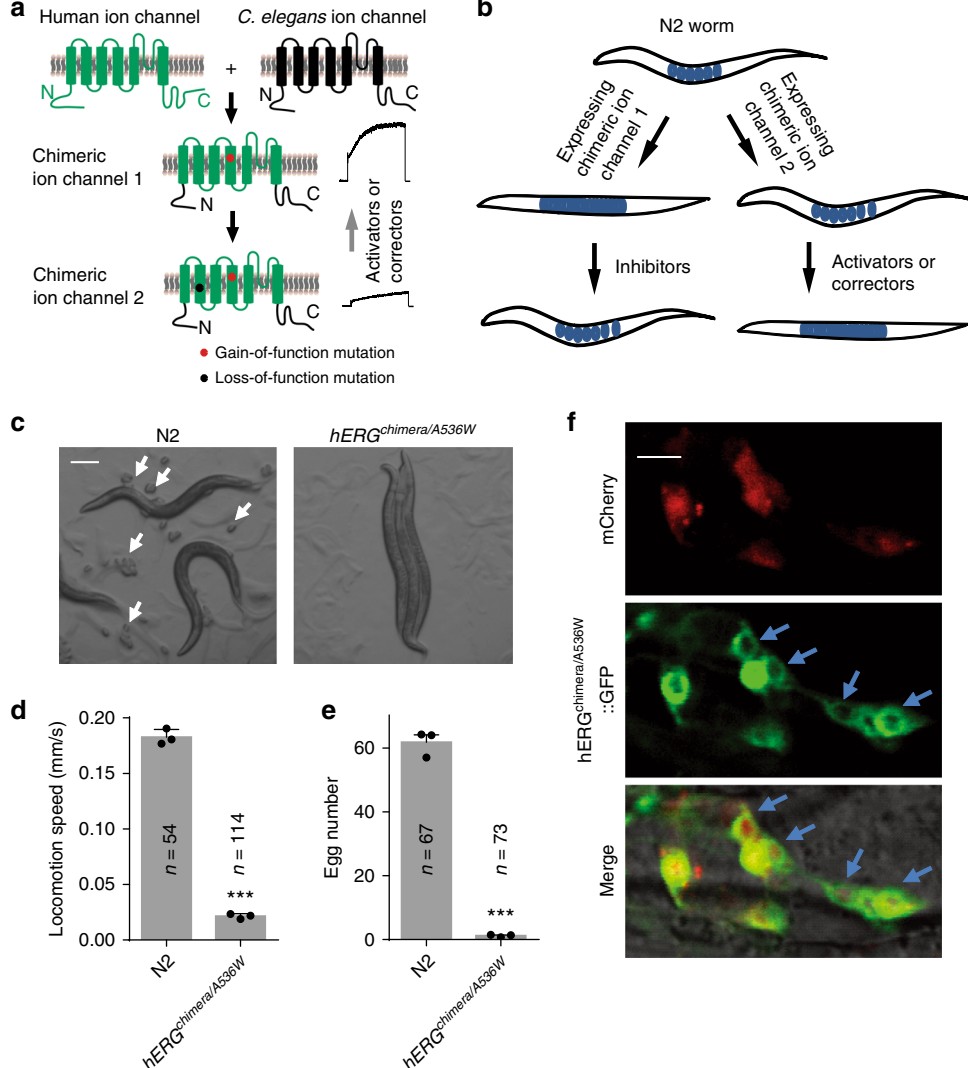

**Fig. 1** *C. elegans* models of channelopathies show behavioral defects. **a** Schematic illumination of chimeric ion channels. The chimeric ion channels consist of transmembrane domains and their connecting loops of a human ion channel, and N-/C-terminus of its *C. elegans* homolog. The gain-of-function mutation was expected to alter cell excitability and consequently induce behavioral defects in transgenic worms. **b** Schematic illumination of generating animal models of channelopathies and strategy for drug screenings. Transgenic worms expressing chimeric ion channel 1 proteins will exhibit clear behavior defects. Drugs that alleviate the phenotypic defects could be inhibitors of the channel. Transgenic worms expressing chimeric ion channel 2 proteins will show no or weak behavior defects due to the reduced channel function, and drugs aggravate the worms' behavioral defects are potential activators or functional correctors of the channel. **c** Microscopy images of wild-type N2 and *hERG^Chimera/A536W* transgenic worms. Scale bar, 200 μM. White arrows indicate *C. elegans* eggs. **d, e** Locomotive (**d**) and egg-laying (**e**) behaviors of of young adult wild-type N2 and *hERG^Chimera/A536W* transgenic worms. **f** Fluorescent images of worms expressing both mCherry and *hERG^chimera/A536W*::GFP proteins that were driven by the *unc-103* promoter. Blue arrows indicate head neurons. Scale bar, 10 μm. All data shown are mean ± s.e.m. ***$P < 0.001$ (Student's *t*-tests for **d, e**)

*hERG^chimera/A536W* worms were ameliorated by individually adding 10 μM well-known hERG blockers, and these effects were markedly enhanced when the *acs-20* null mutation was introduced in the transgenic worms (Supplementary Fig. 2). This result indicates that drugs could be efficiently delivered to their targets, and modified human ion channels expressed in the transgenic worms are functional and the cause of behavioral defects.

**A small-molecule screen identifies a new hERG trafficking inhibitor**. Next, we examined whether *acs-20;hERG^chimera/A536W* worms could be used for identification of hERG K⁺ channel modulators. To identify unknown hERG inhibitors, we screened about 4000 small molecules for their suppression of behavioral

defects in *acs-20;hERG^chimera/A536W* worms (Fig. 2a). Dozens of compounds were found to alleviate the phenotypic defects to some extents. Among them, alphitolic acid (ALA) clearly restored locomotion and egg-laying behaviors of the tested worms (Fig. 2b). Additional test using HEK293T cells expressing wild-type hERG K⁺ channels showed that acute treatment with either 1 or 10 μM ALA did not change the current density (Fig. 2c), suggesting that ALA did not act directly on hERG K⁺ channel permeation. By contrast, long-term (24–32 h) treatment with ALA led to reduced current density of this channel in a dose-dependent manner. The median inhibition concentration ($IC_{50}$) was 8.2 μM, as determined by analyzing changes in deactivating tail currents at the membrane potential of −140 mV (Fig. 2d–f). Western blotting data showed that hERG proteins appeared as two bands corresponding to the endoplasmic reticulum (ER)-

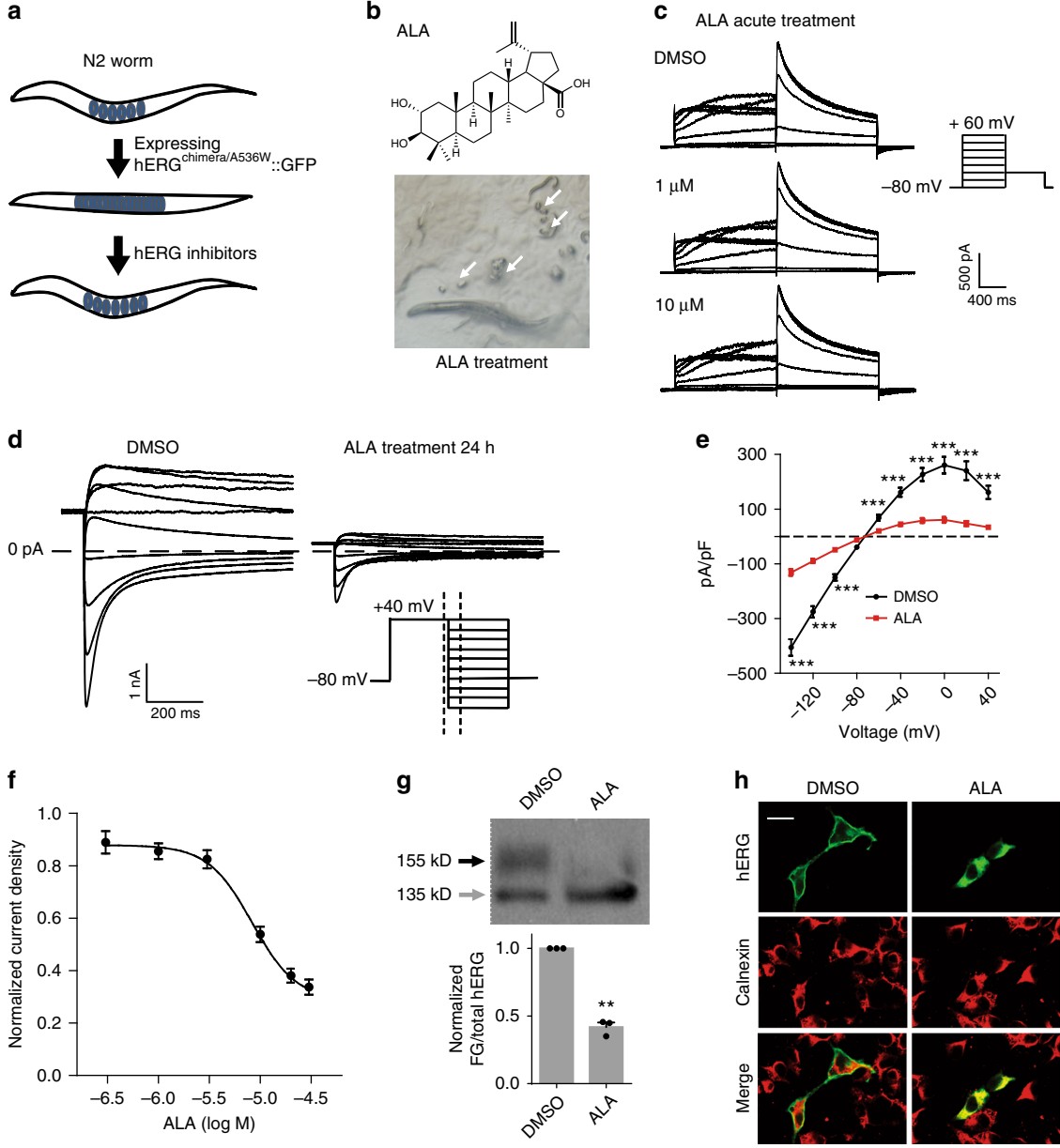

**Fig. 2** A small-molecule screen in *C. elegans* identifies a novel hERG trafficking inhibitor. **a** Schematic illumination of the screening assay. **b** Molecular formula of alphitolic acid (ALA, upper) and microscopic images of *acs-20;hERG*$^{chimera/A536W}$ transgenic worms in the presence of 20 μM ALA in cultivating plates (lower). White arrows indicate eggs. **c** Representative whole-cell currents of HEK293T cells expressing wild-type hERG channels before or after treated with 1 or 10 μM ALA. $n = 6$ cells. **d, e** Representative whole-cell currents and I/V curves of wild-type hERG channels expressed in HEK293T cells treated with DMSO ($n = 19$) or 20 μM ALA ($n = 25$) for 24 h. The protocol is shown in the lower right of **d**, and traces at the time course between two dashed lines are shown. **f** Dose-dependent effects of long-term ALA treatment on the current densities of wild-type hERG channels recorded at a membrane potential of −140 mV. $n \geq 20$ cells per dose were recorded and analyzed. **g** Western blot analysis of hERG proteins expressed in HEK293T cells treated with DMSO or 20 μM ALA for 24 h (top), and quantitantive analysis of the ratio of fully glycosylated (FG, 155 kD) to total hERG proteins (down). Black and gray arrows indicate 155 kD and 135 kD bands of hERG proteins, respectively. **h** Immunostaining of hERG and ER marker protein Calnexin in HEK293T cells treated with DMSO or 20 μM ALA for 24 h. Scale bar: 20 μm. Data shown are mean ± s.e.m. **$P < 0.01$,***$P < 0.001$ (Student's *t*-tests for **e**, **g**). All experiments were performed at least three times

resident (with core glycosylation, ~135 kD) and post-ER (with fully glycosylation, ~155 kD) forms (Fig. 2g). Long-term treatment with ALA reduced the hERG protein levels of the 155 kD, but not the 135 kD band (Fig. 2g). Furthermore, immunostaining of HEK293T cells expressing hERG K$^+$ channels also showed that ALA inhibited protein trafficking of hERG K$^+$ channels (Fig. 2h). These results suggest that ALA is a novel hERG protein trafficking inhibitor. Together, screening assays in worms expressing hERG K$^+$ channels could identify new hERG inhibitors.

**Identification of functional correctors of LQTS2 mutant channels**. The majority of LQTS2-related mutations suppress the $I_{Kr}$ currents by retaining hERG K$^+$ channels in the ER[25,26]. Pharmacological correction of trafficking defects thus represents a promising therapeutic strategy for LQTS[27,28,33,34]. In order to generate an animal model of LQTS, a trafficking-defective LQTS2 mutation A561V[27,35] was introduced to the hERG$^{chimera/A536W}$ protein (Fig. 3a). As expected, the transgenic worms *hERG*$^{chimera/A536W/A561V}$ (expressing hERG$^{chimera/A536W/A561V}$::

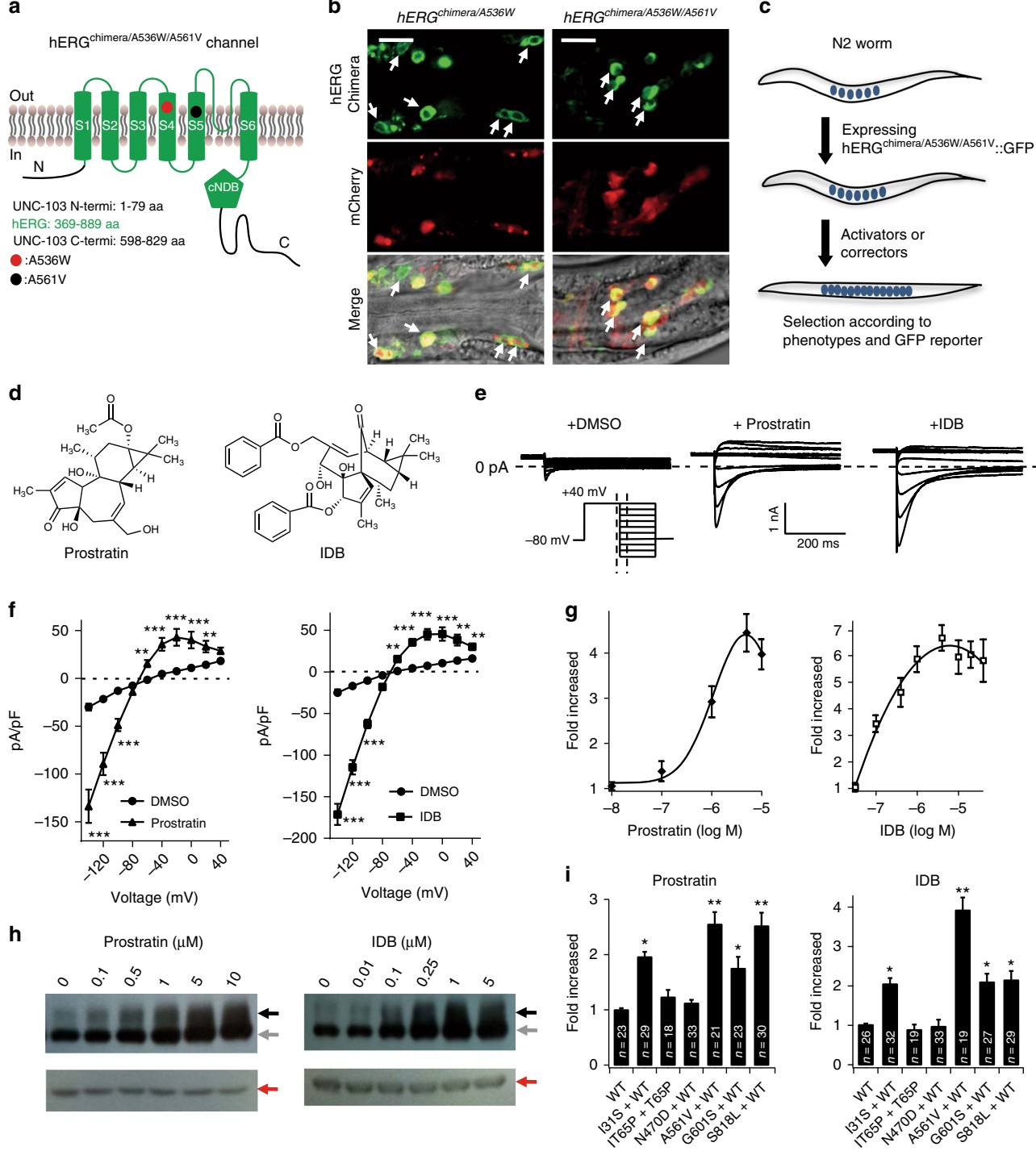

**Fig. 3** Small-molecule screens identify functional correctors of hERG$^{A561V}$. **a** Schematic illumination of construction of hERG$^{chimera/A536W/A561V}$. Mutations A536W and A561V are indicated by a red and a black circle, respectively. aa amino acids. **b** Localization of hERG$^{chimera/A536W}$::GFP and hERG$^{chimera/A536W/A561V}$::GFP expressed in *C. elegans* neurons. White arrows indicate head neurons. Scale bar, 10 μm. **c** Schematic illumination of screening strategy for the discovery of functional correctors. Briefly, we first searched for compounds that caused locomotive and egg-laying defects, and then the candidate compounds were further selected according to surface expression of hERG$^{chimera/A536W/A561V}$::GFP proteins. **d** Molecular formulas of Prostratin and IDB. **e, f** Representative whole-cell currents (**e**) and *I/V* curves (**f**) of hERG$^{WT}$-hERG$^{A561V}$ channels expressed in HEK293T cells treated with DMSO ($n = 26$), 3 μM Prostratin ($n = 21$) or 2 μM IDB ($n = 25$). **g** Dose-dependent effects of Prostratin and IDB on the current density of hERG$^{WT}$-hERG$^{A561V}$ channels recorded at a membrane potential of −140 mV. About 20 cells per dose were recorded and analyzed. **h** Effects of Prostratin and IDB on the protein trafficking of hERG$^{WT}$-hERG$^{A561V}$ channels. Black, gray, and red arrows indicate 155 kD, 135 kD bands of hERG proteins, and tubulin, respectively. **i** Effects of Prostratin and IDB on the current densities of LQTS-related hERG mutant channels. The ratio of hERG$^{WT}$/hERG$^{A561V}$ was 2:1 for **e**, **f**, **g**, and 1:1 for **h**, **I**. The ratio of hERG$^{WT}$/other hERG mutant was 1:1. All experiments were performed at least three times. Data shown are mean ± s.e.m. *$P < 0.05$, **$P < 0.01$, ***$P < 0.001$ (Student's *t*-tests for **f**; one-way ANOVA Dunnett's test for **i**)

GFP proteins) showed GFP reporters in the neuronal cytoplasm (Fig. 3b), indicating mistrafficking of these channel proteins. Consistently, $hERG^{chimera/A536W/A561V}$ worms exhibited nearly normal behaviors (Supplementary Movies 1–3).

The $hERG^{A561V}$ $K^+$ channels could be functional if they reach to the plasma membrane[27]. We hypothesized that if a compound promotes the surface expression of $hERG^{chimera/A536W/A561V}$:: GFP, the transgenic worms would show locomotion and egg-laying defects. We thus generated $acs-20;hERG^{chimera/A536W/A561V}$ worms by crossing $acs-20$ null mutant worms with $hERG^{chimera/A561V}$, and screened for novel functional correctors of $hERG^{chimera/A536W/A561V}$ according to the strategy depicted in the Fig. 3c. Among 10,600 compounds screened at a final concentration of 20 μM, we identified six hits, among which two compounds Prostratin and IDB (Fig. 3d and Supplementary Fig. 3) were known as PKC activators.

The A561V mutation is known to cause strong dominant-negative suppression of wild-type hERG ($hERG^{WT}$) protein trafficking, resulting in reduced channel function at the surface[35,36]. Strikingly, either Prostratin or IDB treatment enhanced the current density of hERG $K^+$ channels in a dose-dependent pattern when we co-expressed $hERG^{WT}$ and $hERG^{A561V}$ at a ratio of 2:1 (Fig. 3e–g). The half maximal effective concentrations (EC50) of Prostratin and IDB were 0.95 μM and 0.042 μM, respectively (Fig. 3g). We then examined whether these compounds affect protein trafficking of hERG mutant channels. In this assay, we co-expressed $hERG^{WT}$ and mutated hERG $K^+$ channels at a ratio of 1:1 because hERG mutations are heterozygous in most LQTS2 patients. Western blotting experiments showed that both Prostratin and IDB promoted trafficking of $hERG^{WT}$-$hERG^{A561V}$ channels in a dose-dependent pattern, as reflected by the elevated 155 kD hERG proteins (Fig. 3h and Supplementary Fig. 4), which distributed outside the ER. In addition to the A561 site, hERG $K^+$ channels with other LQTS-related mutations such as I31S, G601S, and S818L were found to exhibit enhanced function after treatment with 3 μM Prostratin or 2 μM IDB, while the channels with T65P, N470D were not affected by the drug administration (Fig. 3i). Moreover, neither of these two compounds influenced the function and protein trafficking of $hERG^{WT}$ channels (Supplementary Fig. 5). Together, Prostratin and IDB specifically promote the function of some trafficking-defective hERG mutant channels.

**Prostratin and IDB restore the function of LQTS2 mutant channels through PKCε signaling**. Both Prostratin and IDB are PKC activators. Prostratin is non-selective, and IDB is specific for the PKCε. We then examined whether the rescue of trafficking defects by Prostratin and IDB results from activation of PKCε. We found that downregulation of PKCε, but not PKCβ, abolished the effect of IDB and Prostratin on the current density (Fig. 4a–c and Supplementary Fig. 6) and protein trafficking of $hERG^{WT}$-$hERG^{A561V}$ channels (Fig. 4d). Furthermore, over-expression of PKCε enhanced current densities and promoted protein trafficking of $hERG^{WT}$-$hERG^{A561V}$ channels (Fig. 4e, f). Together, these results demonstrate that Prostratin and IDB recover trafficking of hERG mutant channels through the PKCε signaling.

Previous studies[37] have shown that hERG contains 18 putative PKC-dependent phosphorylation sites (Fig. 5a). To explore whether the action of Prostratin and IDB depends on phosphorylation of hERG mutant $K^+$ channels, 17 out of 18 PKC phosphorylation sites (except the T74 site, which prevents functional expression of hERG $K^+$ channel[37]) of the channel were mutated to alanine to block the phosphorylation of these sites. We found that these mutations largely abolished the effects of IDB and Prostratin on correcting the functional defects in LQTS2

mutant channels (Fig. 5b). To further explore which phosphorylation site is required for the function of IDB and Prostratin, we generated three mutated hERG $K^+$ channels with a lack of PKC phosphorylation sites in the N-terminus, the transmembrane domains, or the C-terminus. We found that only blockage of phosphorylation sites in the transmembrane domains diminished the function of IDB and Prostratin (Fig. 5c–e). Further studies showed that a phosphorylation site of hERG $K^+$ channels at S606 (Fig. 5f), but not S636 and T670 (Supplementary Fig. 7), was required for the function of IDB and Prostratin. We next mutated S606 to E606 to mimic the phosphorylation of hERG $K^+$ channels, and found that the mutation S606E in $hERG^{A561V}$ markedly increased the current density of $hERG^{WT}$-$hERG^{A561V}$ (Fig. 5g), suggesting that the phosphorylation of this site promotes the function of the LQTS2 mutant channel. Taken together, our results suggest that IDB and Prostratin activate the PKCε signaling and consequently phosphorylate S606 at the pore region of the channel to correct the defective function of LQTS2 mutant channels.

**Identified compounds restore electrophysiology of cardiomyocytes carrying a LQTS2 mutation**. The hERG $K^+$ channel plays a major regulatory role in the re-polarization phase of cardiac action potentials. Human-induced pluripotent stem cell-derived cardiomyocytes (hiPSC-CMs) from LQTS2 patients (with a heterozygous A561V mutation in hERG) showed diminished $I_{Kr}$ currents and prolonged re-polarization durations[27]. We then generated a heterozygous A561V mutation in hiPSCs by CRISPR/Cas9-mediated genome editing (Fig. 6a and Supplementary Fig. 8a), and differentiated these hiPSCs into cardiomyocytes (named $hiPSC^{A561V}$-CMs). The cardiac ontogeny of both control hiPSC-CMs and $hiPSC^{A561V}$-CMs was confirmed by analyses of cardiac cell-specific transcriptional, structural, and functional markers (Fig. 6b, c).

We found that endogenous $I_{Kr}$ currents, isolated by the hERG blocker E-4031, in $hiPSC^{A561V}$-CMs were significantly decreased in comparison with those in control hiPSC-CMs (Fig. 6d, e). In consistence with this result, immunostaining of $hiPSC^{A561V}$-CMs showed that $hERG^{A561V}$ proteins accumulated in the cytoplasm (Supplementary Fig. 8b). Concurrently, the $APD_{90}$ (action potential duration at 90% of full re-polarization) of atrial-like and ventricular-like $hiPSC^{A561V}$-CMs was about two times longer than that of control hiPSC-CMs (Fig. 6f, g). Furthermore, action potentials in 5 out of 22 $hiPSC^{A561V}$-CMs exhibited early after-depolarization (EAD), the harbinger of cardiac arrhythmias in LQTS, while none of action potentials in control hiPSC-CMs ($n = 25$) showed EADs (Supplementary Fig. 8c, d). Thus, the electrophysiological phenotypes in $hiPSC^{A561V}$-CMs were similar to those in LQTS2 patient-specific hiPSC-CMs[27,38].

We then explored whether Prostratin and IDB could correct the electrophysiological defects of $hiPSC^{A561V}$-CMs. We found that long-term (48 h) administration of Prostratin (3 μM) or IDB (2 μM) markedly enhanced the $I_{Kr}$ currents in $hiPSC^{A561V}$-CMs (Fig. 6d, e). Immunostaining results suggested that Prostratin and IDB ameliorated the trafficking defects of $hERG^{A561V}$ proteins in $hiPSC^{A561V}$-CMs (Supplementary Fig. 8b). More importantly, these two compounds shortened the action potential duration of $hiPSC^{A561V}$-CMs to a level comparable with that of control hiPSC-CMs (Fig. 6f, g) and prevented the occurrence of EADs (Supplementary Fig. 8c, d). By contrast, these two compounds did not affect wild-type $I_{kr}$ currents and the duration of action potentials in control hiPSC-CMs (Supplementary Fig. 9a–d). We further examined the effect of Prostratin and IDB on the transcription of hERG splice variants and other channels expressed in $hiPSC^{A561V}$-CMs, and found that administration

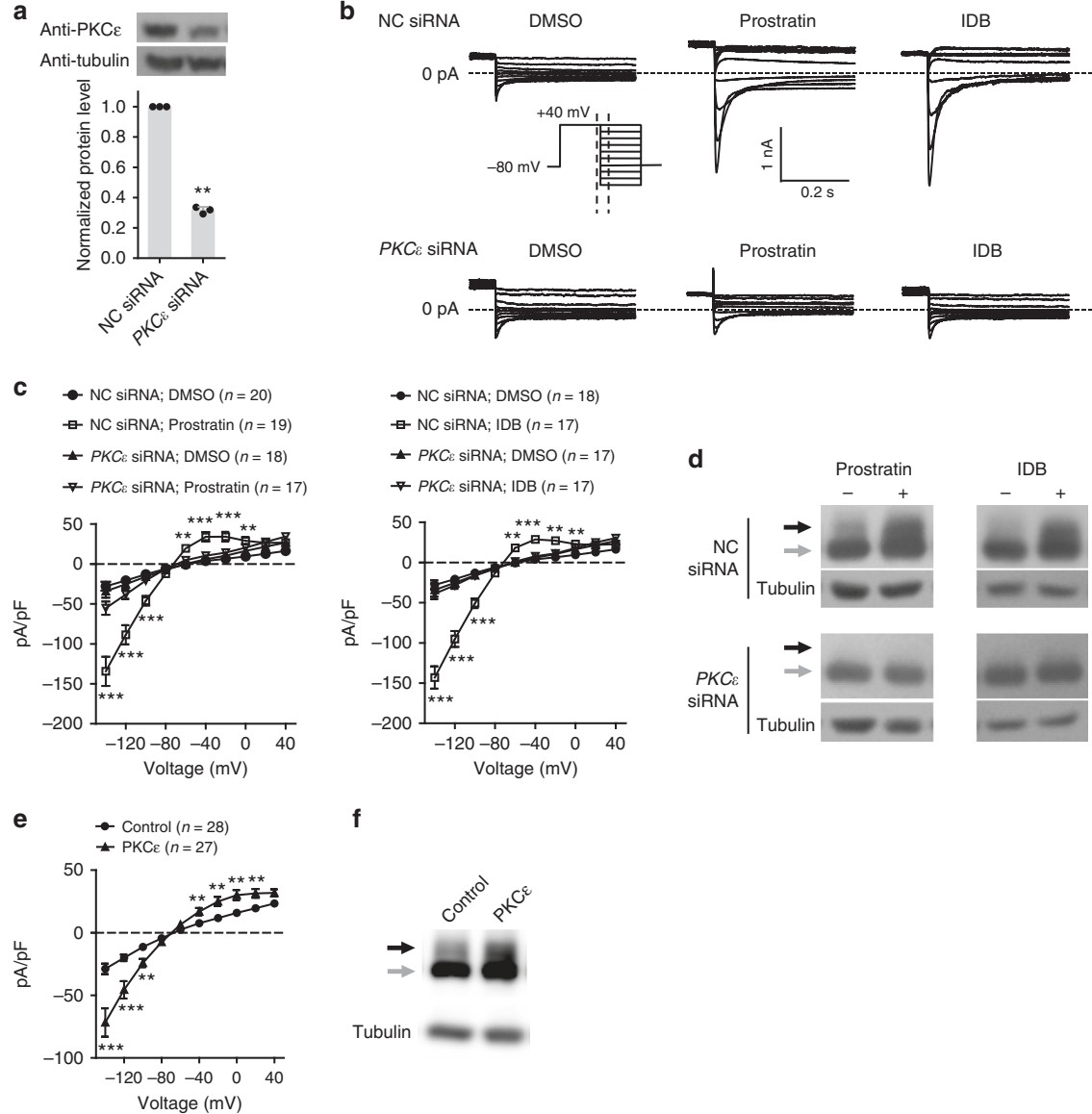

**Fig. 4** The action of Prostratin and IDB requires activation of PKCε signaling. **a** The efficiency of siRNAs targeting *PKCε*. A scrambled siRNA was used as a negative control (NC). **b**, **c** Effects of Prostrain and IDB on whole-cell currents (**b**) and current densities (**c**) of hERG$^{WT}$-hERG$^{A561V}$ channels expressed in HEK293T cells treated with control or *PKCε* siRNAs. **d** Effects of Prostrain and IDB on protein trafficking of hERG$^{WT}$-hERG$^{A561V}$ channels expressed in HEK293T cells treated with control or *PKCε* siRNAs. **e**, **f** Effects of PKCε over-expression on current densities (**e**) and protein trafficking (**f**) of hERG$^{WT}$-hERG$^{A561V}$ channels expressed in HEK293T cells. Black and gray arrows indicate 155 kD and 135 kD bands of hERG proteins, respectively. The ratio of hERG$^{WT}$/hERG$^{A561V}$ was 2:1 for **b**, **c**, **e**, and 1:1 for **d**, **f**. All experiments were performed at least three times. Data shown are mean ± s.e.m. **$P < 0.01$, ***$P < 0.001$ (one-way ANOVA Dunnett's test for **c**, Student's *t*-tests for **a**, **e**)

of Prostratin and IDB marginally influenced the expression of these examined ion channels (Supplementary Fig. 9e). Together, these data demonstrate that Prostratin and IDB correct electrophysiological abnormalities in LQTS2-specific hiPSC-CMs. Thus, small-molecule compounds similar to Prostratin and IDB are potential drugs for treating patients with LQTS2.

## Discussion

The discovery of new ion channel drugs could be greatly facilitated by the availability of an economic, reliable, robust, and high-throughput in vivo assay. In this study, we generated *C. elegans* models of channelopathies by expressing disease-related hERG channels in the worms, and developed phenotype-based screening assays by which we successfully identified some novel channel

modulators, illustrating the usefulness of this in vivo high-throughput screening method.

Theoretically, inherited mutations could result in dysfunctions of ion channels by the following mechanisms: abnormal transcription/translation, deficient assembly or protein trafficking, defective channel gating/kinetics, and altered channel permeability[22,33]. Different disease-related mutant channels may need different strategies to rescue their function. For instance, the function of an ion channel mutant with abnormal transcription, translation, or defective trafficking may not be fully corrected by activators of ion channels. Current drug screenings, however, mainly focus on identification of compounds that modulate the gating and kinetics of ion channels. In this study, we have demonstrated that different screening methods could be designed by introducing specific disease-related channel mutations into *C.*

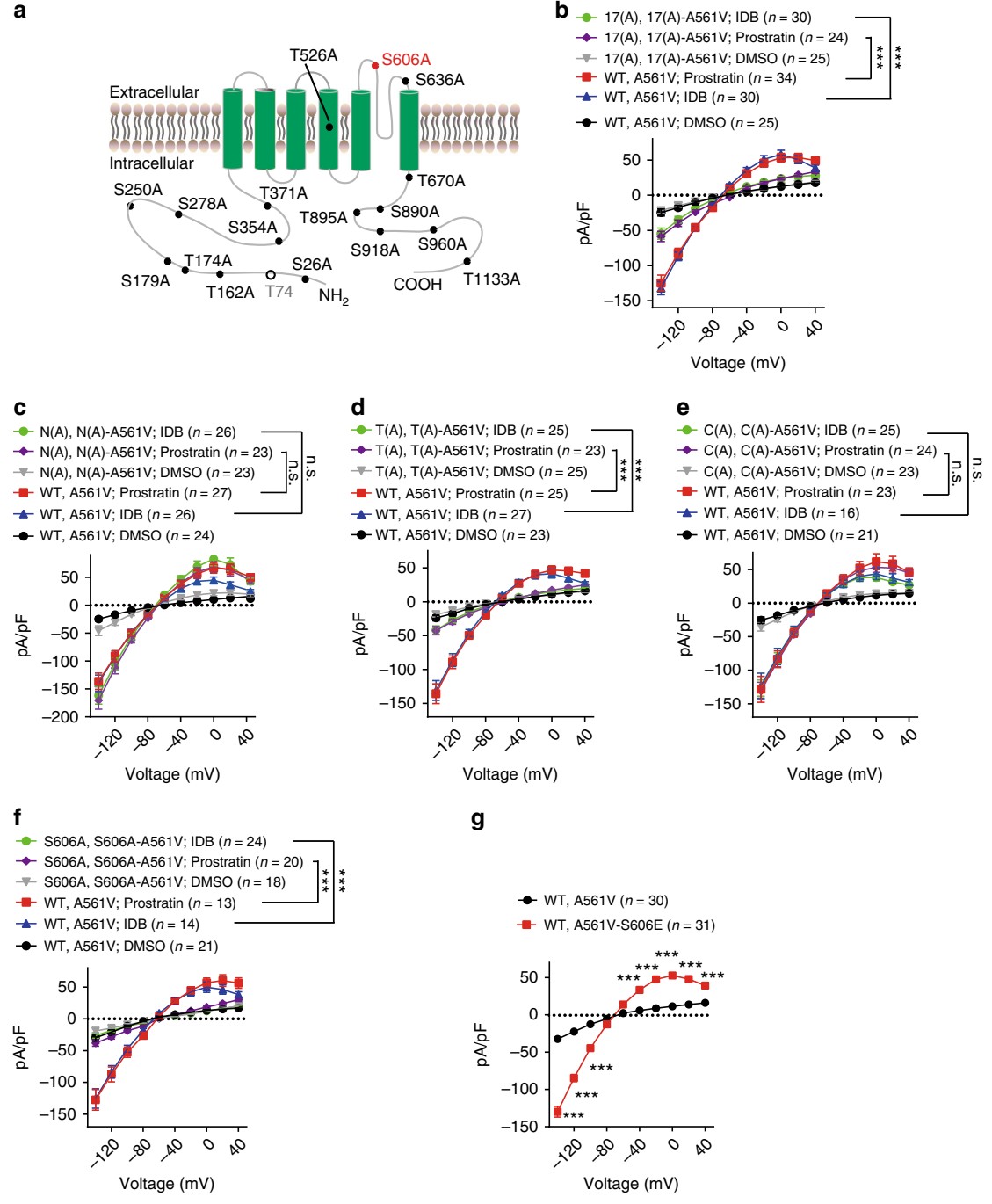

**Fig. 5** The action of Prostratin and IDB depends on the phosphorylation of the S606 at the pore region of the hERG channel. **a** Schematic illumination of putative PKC phosphorylation sites and point mutations in hERG proteins. **b** Effects of Prostratin and IDB on the current densities of hERG$^{WT}$-hERG$^{A561V}$ channels with a lack of 17 PKC phosphorylation sites (indicated as 17(A)). **c–e** Effects of Prostrain and IDB on the current densities of hERG$^{WT}$-hERG$^{A561V}$ channels with a lack of PKC phosphorylation sites (S26A, T162A, T174A, S179A, S250A, S278A, S354A, and T371A) in the N-terminus (indicated as N(A)), PKC phosphorylation sites (T526A, S606A, S636A, and T670A) in the transmembrane domain (indicated as T(A)), or PKC phosphorylation sites (S890A, T895A, S918A, S960A, and T1133A) in the C-terminus (indicated as C(A)). **f** Effects of Prostrain and IDB on the current densities of hERG$^{WT}$-hERG$^{A561V}$ channels with a lack of the S606 PKC phosphorylation site. **g** Effects of the mutation S606E on the current densities of hERG$^{WT}$-hERG$^{A561V}$ channels. The ratio of hERG$^{WT}$/hERG$^{A561V}$ was 2:1. All experiments were performed at least three times. Data shown are mean ± s.e.m. ***$P < 0.001$, n.s., not significant (one-way ANOVA Dunnett's test for **b**, **c**, **d**, **e**, and **f**; Student's $t$ tests for **g**)

*elegans* disease models. Phenotype-base screens in these animal models could discover compounds that correct abnormal channel phenotypes caused by specific mutations.

The cumulative evidence suggests that defective protein biogenesis represents a rather common mechanism underlying many channelopathies, including LQTS, cystic fibrosis, episodic ataxia

type 1, epilepsy, and hearing loss[25,26,33,39–41]. Despite extensive interests in pharmacological correction of defective protein biogenesis of ion channels, few such compounds have been found. A successful example is that a combination administration of CFTR$^{\Delta F508}$ protein trafficking corrector VX-809 and CFTR chloride channel potentiator VX-770 was approved for the

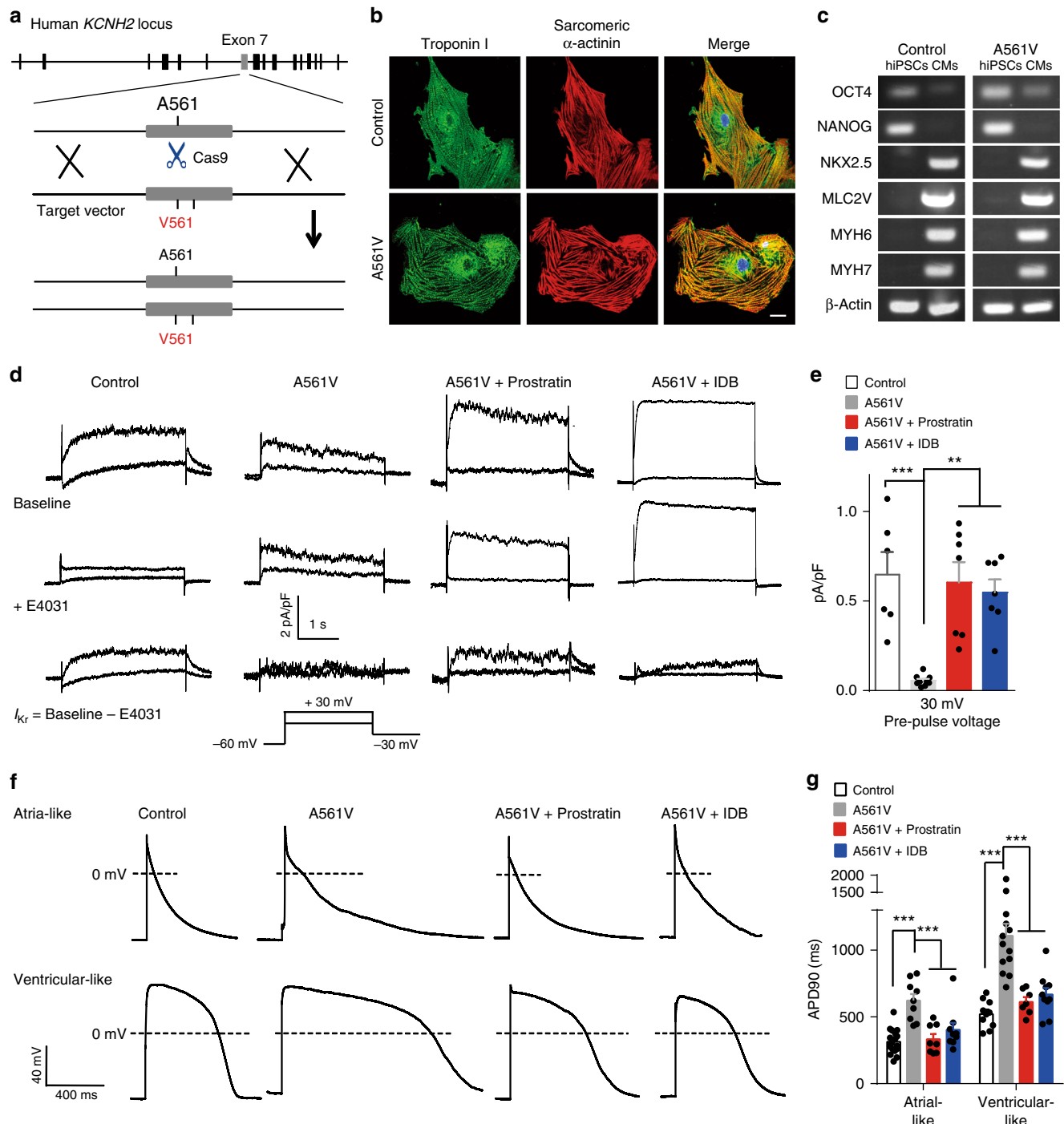

**Fig. 6** Prostratin or IDB corrects electrophysiological abnormality of hiPSC[A561V]-CMs. **a** Schematic illumination of CRISPR/Cas9-mediated genome editing in *KCNH2* (encoding the hERG channel) locus. **b** Immunostaining for sarcomeric α-actinin and Troponin I in control hiPSC-CMs and hiPSC[A561V]-CMs. Scale bar: 20 μm. **c** RT-PCR analysis of the cardiac-specific transcription factors (*NKX2.5*) and structural genes (*MLC2V, MYH6,* and *MYH7*) in control hiPSC-CM and hiPSC[A561V]-CMs. **d** Representative $I_{Kr}$ currents. **e** Tail current densities of $I_{Kr}$. $n = 6, 9, 7,$ and 7 cells for control hiPSC-CMs, hiPSC[A561V]-CMs, Prostratin-treated hiPSC[A561V]-CMs, and IDB treated hiPSC[A561V]-CMs, respectively. **f** Representative action potentials. **g** APD90 of action potentials. $n = 15, 9, 8,$ and 9 atrial-like cells for control hiPSC-CMs, hiPSC[A561V]-CMs, hiPSC[A561V]-CMs + Prostratin, and hiPSC[A561V]-CMs + IDB, respectively. $n = 10, 13, 7,$ and 10 ventricular-like cells for control hiPSC-CMs, hiPSC[A561V]-CMs, hiPSC[A561V]-CMs + Prostratin, and hiPSC[A561V]-CMs + IDB, respectively. Data shown are mean ± s.e.m. **$P < 0.01$, ***$P < 0.001$ (one-way ANOVA Dunnett's test for **e**, **g**). All experiments were repeated at least three times

treatment of cystic fibrosis[42]. Some hERG blockers have been reported to promote protein trafficking of LQTS mutant channels[26,43], however, the blockage action of drugs may diminish the current conducted by rescued hERG K⁺ channels, consequently nullify their therapeutic potential. In this study, our phenotype-based screening assays in *C. elegans* model of LQTS led to identification of compounds that not only promote protein trafficking of LQTS mutant channels, but also correct the electrophysiological abnormality of cardiomyocytes carrying a LQTS2 mutation. Thus, the *C. elegans* models of LQTS provide useful

in vivo systems for reliable identification of functional correctors of LQTS2 mutations.

Our screening has identified two hERG modulators Prostratin and IDB that are PKCε activators. The PKC signaling is known to act on cardiac sodium channels, $K_{ATP}$ channel, KCNQ1, and hERG in regulating the re-polarization of cardiac action potentials[44–46]. We found that these two compounds restored the normal function of LQTS-related mutant hERG $K^+$ channels by correcting the biogenesis defect, without affecting wild-type hERG and cardiomyocyte physiology. The PKCε may be a specific isoform of PKC that targets biogenic processing of hERG through phosphorylating the pore region of the channels. The selective and effective action of Prostratin and IDB is particularly important for their potential clinical applications. Prostratin is a natural product found in the bark of the mamala tree, and it has a potential in treatment of AIDS by specifically targeting the latent HIV reservoirs[47,48]. Whether Prostratin and IDB are effective for treating LQTS patients remains to be further studied.

In summary, we demonstrate that *C. elegans* models of channelopathies are excellent systems for the discovery of ion channel drugs. The robustness of this method is evidenced by the fact that we have successfully identified six compounds as functional correctors of hERG mutant channels in this study, while only about ten hERG activators have been reported previously[49]. Since most types of ion channels are well-conserved between *C. elegans* and human, particularly in the transmembrane domains where mutations for channelopathies are often found, we proposed that *C. elegans* models of channelopathies could be generated for other channels besides the hERG $K^+$ channel. Phenotypes of *C. elegans* channelopathy models are easy to be quantified, and hence a reasonable high-throughput screen could be done in a regular ion channel lab. Small-molecule screens in this study were done manually with about 250 drugs per person per day. Several high-throughput behavioral analysis platforms for *C. elegans* have been developed for different purposes[50–52]. With proper automation in behavioral screening, we expect that a high-throughput screening system based on our method will become an efficient tool for drug discovery for channelopathies.

## Methods

**Worm strains.** *C. elegans* strains were cultured on Nematode Growth Medium (NGM) at 20 °C. The wild-type N2 Bristol strain and the *acs-20(tm3232)* strain were obtained from Caenorhabditis Genetics Center and National BioResource Project, respectively. *hERG*chimera/A536W and *hERG*chimera/A536W/A561V transgenic worms were generated by expressing P$_{unc-103}$::hERGchimera/A536W::gfp and P$_{unc-103}$::hERGchimera/A536W/A561V::gfp plasmids in N2 worms, respectively. The P$_{unc-103}$::mcherry plasmid was co-injected with the above plasmids to express mCherry as a cytoplasmic protein marker in these transgenic worms. The transgenic worms were integrated by the UV-TMP method[53] and outcrossed for three times.

**Molecular biology.** The hERGchimera was constructed by fusing UNC-103 (isoform a) N-terminus (1–79th amino acids) with hERG transmembrane/cNBD domains (369–889th amino acids) and UNC-103 C-terminus (599–829th amino acids). PCR product of *HA-HERG* cDNA was subcloned into the pCI-neo vector to construct plasmids that could be expressed in mammalian cell lines. Mutations were generated by site-directed mutagenesis. All constructs were confirmed by DNA sequencing.

**Immunoblot analysis.** Western blot analyses were performed as described below. Briefly, 16 h after transfection, HEK293T cells were treated with drugs and on the next day cells were lysed in the NP-40 lysis buffer (1% NP-40, 150 mM NaCl, 1 mM NaF, 50 mM Tris, pH 7.6) supplemented with protease inhibitor mixture (05892791001, Roche). After centrifugation at $12,000 \times g$ at 4 °C for 10 min, the supernatants were collected for Western blot analyses (full blots found in Supplementary Fig. 10). The primary antibodies used in this study were rat anti-HA (11867423001, Roche) with a dilution 1:3000, rabbit anti-PKCε (sc-214, Santa Cruz) with a dilution 1:1000, mouse anti-β-Tubulin (M30109, Abmart) with a dilution 1:3000.

**Immunocytochemistry.** The HEK293T cells or hiPSC-CMs were fixed with 4% paraformaldehyde for 20 min. After washed for three times by PBS, the fixed cells were permeabilized with 0.1% Triton X-100 for 10 min and then were blocked with 10% bovine serum albumin for 1 h. For immunocytochemistry of HEK293T cells, mouse anti-HA (M20003, Abmart; 1:400), and rabbit anti-Calnexin were used as primary antibodies (ab22595, Abcam; 1:400); for immunocytochemistry of hiPSC-CMs, rabbit anti-troponin I (sc-15368, Santa Cruz; 1:500) or anti-hERG (ab196301, Abcam; 1:200) and mouse anti-sarcomeric α-actinin (A7811, Sigma; 1:600) were used as primary antibodies. Anti-mouse Alexa Fluor 594 and anti-rabbit Alexa Fluor 488-conjugated secondary antibodies were used for immunofluorescence detection.

**Behavioral assays.** Synchronized young adult worms were used for behavioral assays. For egg-laying assays, individual late-L4 hermaphrodite was picked on a NGM plate with *E. coli* strain OP50 and was picked away 24 h later. Total eggs and progenies in the plate were counted. The locomotion speed was measured on the NGM plate without OP50. About 25 young adult worms were picked on the plate and allowed to recover for 1 min, and then their spontaneous movement was recorded for 1 min. The locomotion speed was calculated by using wrMTrck software. For male's copulatory spicule protraction assays, virgin L4 males were isolated on NGM plates seeded with OP50. After 24 h, male spicule protraction was scored if at least one spicule partially extended from the cloaca. At least 40 worms per strain were tested in each experiment. The investigators were blinded to the genotypies during *C. elegans* behavioral assays.

**Small-molecule screen.** Chemical libraries used in this study include the Prestwick Chemical Library, the Spectrum Collection, the Natural Products Library, the ICCB Known Bioactives Library, the IB screening Library, the NIH Clinical Collection, the FDA Approved Drug Library, the SIGMA LOPAC, and the AnalytiCon Discovery. 48-well screening plates were prepared by adding 500 μl NGM containing 1 mM IPTG and 25 μg ml$^{-1}$ carbenicillin in each well. A mixture of HT115 *E. coli* expressing double-strand RNAs targeting *ifd-2* and *c15c7.5* were then seeded on NGM plates and allowed to grow for 2 days. Small molecules dissolved in DMSO were then added to NGM plates with a final concentration of 20 μM. After 4 h, synchronized L4 *acs-20;hERG*chimera/A536W or *acs-20;hERG*chimera/A536W/A561V worms were rinsed from their cultivating plates and deposited into each well of these screening plates. Phenotypes of tested worms were examined 36 h later.

**Electrophysiology.** Whole-cell patch-clamp was performed by using Axon 200 B. For recording currents in HEK293T cells expressing hERG channels, pipette solution consisted of (in mM) 135 KCl, 10 EGTA, 1 $MgCl_2$, 5 Mg-ATP and 10 HEPES (pH 7.2 with KOH), and bath solution consisted of (in mM) 130 NaCl, 5 KCl, 10 HEPES, 1 $MgCl_2$, 10 glucose, and 2 $CaCl_2$ (pH 7.4 with NaOH). Recording was performed at room temperature by using a standard protocol that membrane potential was first pre-depolarized to +40 mV to inactivate the channel, and then repolarized voltage from −140 mV to +40 mV (in 20 mV increments) to obtain tail currents.

To measure $I_{Kr}$ currents and cardiac action potentials, hiPSC-CMs were cultured on 0.1% gelatin coated coverslips in a 24-well plate. 3–8 days after seeding, hiPSC-CMs were treated with 2 μM IDB (sc-202663, Santa Cruz) or 3 μM Prostratin (P0077, Sigma) for 48 h. The $I_{Kr}$ currents of hiPSC-CMs were isolated by application of hERG blocker E-4031 (M5060, Sigma). $I_{Kr}$ was measured using a protocol that initially depolarizing the membrane potential from −60 mV to + 60 mV in 30 mV increments, then repolarizing to −30 mV to elicit tail current. The composition of the pipette solution was (in mM): 120 KCl, 1 $MgCl_2$, 3 Mg-ATP, 10 HEPES, and 10 EGTA (pH 7.2), and bath solution was (in mM): 126 choline chloride, 5.4 KCl, 1.8 $CaCl_2$, 1 $MgCl_2$, 10 HEPES, and 10 glucose (pH 7.4). Nifedipine (5 μM; N7634, Sigma) and chromanol (10 μM; C2615, Sigma) were added in the bath solution to suppress potential interference of $I_{Ca}$ and $I_{Ks}$, respectively.

Action potentials were elicited by stimulating cardiomyocytes with depolarizing 10 ms-impulses from 0 pA to 40 pA by 4-pA increments. The frequency of stimulus for the protocol was about 0.5 Hz. The bath solution contained (in mM) 140 NaCl, 5.4 KCl, 1.8 $CaCl_2$, 1 $MgCl_2$, 10 HEPES, and 10 glucose, pH 7.4. The pipette solution contained (in mM) 120 KCl, 1 $MgCl_2$, 3 Mg-ATP, 10 HEPES, and 10 EGTA, pH 7.2.

**Generation of heterozygous hERG$^{A561V}$ hiPSC clones.** Generation of heterozygous hERG$^{A561V}$ hiPSC clones was carried out by following a similar protocol described previously[54]. Briefly, wild-type 1016 hiPSC (HSCI iPS Core, Harvard) were grown in feeder-free adherent culture in chemically defined mTeSR1 (STEMCELL Technologies) supplemented with penicillin and streptomycin. Plates were precoated with Geltrex matrix (Invitrogen). The cells were disassociated into single cells with Accutase (Invitrogen), and 10 million cells were electroporated with a mix of 30 mg of the CRISPR plasmid and 30 mg of the donor plasmid in a single cuvette (Bio-Rad). The cells were then collected from the culture plate 48–72 h post-electroporation, and resuspended in PBS buffer. Cells expressing green fluorescent markers were collected by FACS (FACSAria II; BD Biosciences) and

replated on 10 cm tissue-culture plates at around 20,000 cells per plate to allow for recovery in growth medium.

Post-FACS, the cells were allowed to recover for a week, and then single colonies were manually picked and replated individually to wells of 96-well plates. Colonies were allowed to grow to near full confluence over the next 7 days, at which point they were split and replica-plated. Genomic DNA of colonies was extracted and genotyping at the CRISPR target site was identified by PCR amplication and DNA sequencing. Clones with confirmed heterozygous mutation allele or wild-type allele were expanded for further experiments.

**Generation of cardiomyocytes from hiPSC.** Generation of cardiomyocytes from hiPSC was performed by following the previously reported method[55] with modifications. Cardioeasy chemically defined cardiac differentiation kit (CA2004500, Cellapy) was used for differentiation hiPSC into cardiomyocytes. Briefly, human hiPSC were splitted using 0.5 mM EDTA PBS solution at 1:10 ratio and cultured with PSCeasy chemically defined medium for 3 days. When reached ~80% confluence, hiPSC was cultured in Induction Medium II for 2 days and then in Induction Medium III. Induction Medium III was changed every other day afterwards. Spontaneously contracting cells were observed from the day 8 after induction. The derived cardiomyocytes were purified with Cardioeasy purification medium at the day 15 after induction.

**Statistical analysis.** Statistical analysis was done by using GraphPad Prism 6.0. Data shown are mean ± s.e.m. For the *C. elegans* behavior data, unpaired Student's *t*-test was used to determine significant differences between samples. For electrophysiological data, one-way ANOVA Dunnett's test was used when comparing multiple groups. Significance levels: *$P < 0.05$, **$P < 0.01$, ***$P < 0.001$. The ALA inhibition curve was fitted with equation: $y(x) = y_{min} + (y_{max} - y_{min})/(1 + 10^{(x-\log IC50)})$. Where $x$ is the log of concentration; $y(x)$ is response, decreasing as $x$ increase; IC50, the median inhibitory concentration. The stimulation curve of Prostratin or IDB on hERG$^{A561V}$ current densities was performed with Gaussian fit. No statistical method was used to predetermine sample size. All experiments were repeated at least for three times.

## Data availability

All data supporting the findings of this study are included in this article (and its Supplementary Information files). Further data are available from the corresponding author upon reasonable request.

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

## Acknowledgements

We thank Mu-ming Poo and Ying Liu for critical reading of the manuscript, Mei-Yu Ruan for technical assistance, the C. elegans Genetics Center (funded by NIH Office of Research Infrastructure Programs P40 OD010440) and S. Mitani (Tokyo Women's Medical University School of Medicine) for providing us strains and National Compound Resource Center for providing us small-molecule libraries. This work was supported by grants to S.-Q.C. (the Strategic Priority Research Program of Chinese Academy of Science, XDBS1020100 and the National Natural Science Foundation of China, 31471149 and 81527901), to F.L. (the National Natural Science Foundation of China, 81422003), and to Q.D. (the National Key R&D Program of China, 2017YFA0102800 and 2017YFA0103700).

## Author contributions

Q.J., K.L. and S.-Q.C. designed all the experiments. Q.J. and K.L. performed most of the experiments. Q.J. generated the *C. elegans* models of channelopathies, performed small-molecule screening assays, and the electrophysiological experiments. K.L. performed the biochemical assays. W.-J.L. conducted the cardiomyocyte differentiation of the hiPSCs. S.L. and Q.J. generated the CRISPR/Cas9-edited hiPSC clones. X.C. performed the electrophysiological recordings of some of the hERG mutant channels expressed in HEK293T cells. J.Y. and X.-J.L. assisted in carrying out the small-molecule screenings. Q.J. and K. L. performed the behavioral assays. F.L. supervised the cardiomyocyte differentiation experiments and analyzed the data. Q.D. supervised the CRISPR/Cas9 editing experiments and analyzed the data. Q.J., K.L., F.L., Q.D. and S.-Q.C. prepared the figures and wrote the manuscript.

## Additional information

**Competing interests:** W.-J.L is a cofounder of Beijing Cellapy Biological Technology Co., Ltd. The remaining authors declare no competing interests.

