## [Peer Review File · Nature Communications]

Reviewers' comments:

Reviewer #1 (Remarks to the Author):

Jiang et al present a beautiful story that starts with the construction of worm models for four different channelopathies, performing drug screens against one of these models (hERG-LQTS2), the successful test of the worm-derived hits in human cells, and the rigorous examination of the mechanism of action of their hits in two cell-based models of the disease. I find the work compelling, exciting, of broad interest and support its publication in Nat Comm. Aside from minor errors in grammar, the manuscript itself is easy to digest and the figures are beautifully presented and easy to absorb. Despite my many comments below, I am very excited about this work/manuscript.

My one major concern is focused around a secondary theme of the paper. Namely, that the hERG model models can be used to test the safety of drugs against inhibition of hERG function, which is a major concern when testing new candidate therapeutics. Because the authors found that ALA, clofilium and E-4031 can inhibit the effects of the GF version of the worm hERG model, they conclude on lines 197 and 198 that their model can be used as an inexpensive hERG safety test. There are several problems with this conclusion. First, the worm is not a human system, and despite making the worm hypersensitive to small molecule penetration, it is not possible to state with any certainty whether a molecule that shows no effects in their worm assay actually got into the worm and can reach a potential target. So, a molecule that fails to suppress the worm hERG GF model may not even be able to reach the nERG target. Second, the hERG GF model is just that, GF. So it is possible that a molecule may be able to suppress WT hERG activity, but for any number of reasons, not be able to suppress a GF type of allele. Third, its not at all clear that the worm hERG GF model phenotype has anything to do with WT ERG activity. The phenotype could be due entirely to neomorphic properties. Hence, suppression by some mechanisms will be relevant to WT ERG activity, while suppression by other mechanisms will not be relevant to WT ERG activity. IMHO, all of this is easily addressed by re-writing parts that deal with safety- either remove that part of the story entirely, or make it clear that its proof-of-principle as a safety model, but a lot more work needs to show that its reliable.

Major/minor Issue: The inclusion of the four models.

First, I found the transition between the description of the various models to be jarring/abrupt. Might want to use, 'We made four different models. First, (model 1). Second, (model 2) etc' so that its clear it's a list.

Second, I didn't understand the need to include the three other models aside from hERG. They seem superfluous. Save them for other papers and tighten up this paper by removing them.

Third, it would have been informative to know more about the expression pattern and biological role of the worm channels that the authors were exploiting.

Fourth, if the corresponding worm gene has a loss-of-function phenotype, it would be good

to know if the human (chimera or not) transgene can rescue that loss-of-function phenotype. Rescue would establish physiological relevance to the modeled human channel. Or maybe rescue doesn't matter so long as the neomorphic/GF phenotypes can be blocked with known inhibitors? Either way, the issue deserves to be addressed in the manuscript.

Minor issues:

1. The writing of the results is well done. Elsewhere could use some help with grammar.

line 39- change 'identify' to 'reveals'

line 53- are the molecular basis

line 56- remains an urgent

line 71- and are known

line 143- R201C, which is found

line 331- different screening methods

line 333- 'that aiming at compounds that' needs to be revised.

line 336- 'Cumulating' needs to be revised.

line 385- '250 drugs per person per day'

2. Lines 57 and 58. That existing drugs were mostly discovered using animal models is a contentious statement and I have a hard time believing that. More literature should be investigated or this statement should be removed/modified.

3. It would be helpful to explain to the reader why chimeric channels were required at all. Did the unmodified human transgenes fail to elicit phenotype?

4. Line 159 etc.

A. It would be good to know whether the assays work in liquid (for HTP analyses) or must everything be done in solid because of the fragility of the acs-20 mutant?

B. Also, it would be good to explicitly state whether acs-20 was actually needed to see suppression.

C. It would be good to present a rationale as to why the existing inhibitors are not sufficient therapies. loss-of-function phenotype

5. Line 174. Well, the GF phenotypes indicate that they have some function, but not necessarily wt function.

Reviewer #2 (Remarks to the Author):

The manuscript describes an experimental approach whereby the model organism *C. elegans* is used to identify ion channel correctors. The method is focused on potassium channel defects associated with cardiac arrhythmia. Identified compounds are validated in HEK and derived cardiomyocytes and are shown to exert their effect on hERG trafficking via PKC ϵ which phosphorylated S606 to promote hERG561V trafficking.

Comments

The use of the *C. elegans* model system does not provide improved clarity or efficiency in identifying new ion channel modulators and correctors. As described, the compounds that are discovered could have just as easily been obtained by HTS electrophysiological approaches for HEK cells (or similar) expressing hERG channels. It is not clear from the data as presented if the discovered compounds impact multiple ion channels. The possibility that site-specific phosphorylation promotes trafficking is intriguing but these data do not further the logic underlying the use of *C. elegans* for drug discovery. Overall, the approach is not well justified. In the end, the study has not advanced the understanding of channel gating or trafficking.

The logic of combining the A536W gain-of-function gating mutation with the A561V trafficking defect is unclear. The identification of Prostratin and IDB as trafficking modulators via PKC phosphorylation is new and somewhat intriguing.

P5. "Given that ion channel inhibitors and activators often target at transmembrane 116 domains of the channels,....". It is assumed from this statement that the authors are referring to pore-blockers? If, so then this is an over-simplification of the literature. Peptide toxins, for instance, are potent ion channel modulators that bind to loops at the membrane periphery.

Figure 1D. Are these channels WT hERG or hERG A526W? Regardless, the utility of AHA to degrade is not clear. Simply removing some channels that have a gain-of-function gating defect would still leave a small population of mutant channels with gating defects.

The rescued K currents in cardiomyocyte A561V cells display perturbed activation and inactivation kinetics compared to control K currents (Figure 6, panel d). These K currents seem to correct the cardiac action potential in subsequent experiments, but is it possible that in this preparation that another channel or splice variant is upregulated?

There are major issues with grammar throughout the manuscript. For instance, first line of the Introduction "Ion channels are molecular basis for cell excitability" is missing a "the" between "are" and "molecular".

Minor

The mutant in figure 1 is described as Kv1.2chimera/R297C but in the text this clone is referred to as KCNQ2chimera/R201C

Reviewer #3 (Remarks to the Author):

This manuscript presents several *C. elegans* animal models as platforms for high-throughput screening for channel inhibitors and activators. The significance is that compounds identified

in such a "native" cellular environment may be more robust and authentic than those identified in screens performed using more reduced heterologous expression systems. For example, *C. elegans* screen for hERG inhibitors, it is argued, could serve as a better platform for detecting drugs in development that might cause drug-induced long QT syndrome (LQTS). A screen identifying chemical correctors of aberrant trafficking could, in principle, lead to drugs to treat the majority of inherited LQT2 cases attributed to hERG trafficking defects. The animal models were created by making chimeras between variants of *C. elegans* UNC-103 and hERG, its human homolog. The chimeras carry the N- and C-termini of UNC-103 and the hERG transmembrane domains, where most interacting compounds are thought to act. Expression of the constructs were driven by the UNC-103 promoter. Overall, the model system approach is clever and could contribute new biological understanding, in addition to its utility as a screen for new drug candidates. However, there were weaknesses in the manuscript, many of which could be addressed.

1. The most intriguing findings were obtained using the heterozygous mutant A561V, a congenital LQT2 mutation that, in heterologous systems, disrupts trafficking. Screens using this mutation identified two compounds, Prostratin and ingenol-3,20-dibenzoate (IDB), that correct the associated mutant behavior in the worms. Their effects in restoring hERG mutant currents expressed in HEK293 cells and on IKr in mutant iPSC-CMs is convincing. It would also be interesting to see the effects of these compounds on WT IKr, considering the risk associated with hERG activators in triggering short QT syndrome, an equally malignant arrhythmia.

2. The effect of these compounds is aptly termed "functional correction" as the behavior is restored and currents are clearly enhanced. It is difficult to know based on the evidence provided whether these compounds promote trafficking or retard degradation, since maturation measurements were not provided. It would be fairly straightforward to do this, with a measurement of the density of the fully-glycosylated 155kD band/total 135+155. This would enhance the mechanistic aspects of the manuscript.

3. The great majority of LQT2 hERG mutations cause trafficking defects when expressed in HEK293 cells, but whether this is the disease phenotype in vivo has yet to be determined in most cases. In this regard it would be a contribution to include immunofluorescence data in Fig. 6 of the hERG signal. If the V561A mutant protein shows accumulation in the cytoplasm and subsequent amelioration with prostratin and IDB, this would not only add mechanistic information to the study but would provide veracity to the assumption from HEK293 cell studies that these mutations may cause trafficking aberrations even in more native-like settings.

4. The case for using *C. elegans* to screen for drugs that block hERG was not very strong. The animals must be heavily manipulated to effectively expose them to the compound, which also seem to be in high concentration. It is hard to see this assay supplanting existing assays that are more sensitive, specific and clearly relevant to drug-induced LQTS.

5. The western blot in Fig. 3 showing absence of maturation by a hERG-inhibiting compound is not convincing as a sole piece of evidence. This should be presented with summary

statistics for upper band maturation (as described above) over several western blots.

6. The way the IKr currents are represented is problematic. Although IKr has inwardly rectifying properties, it is not a classical inward rectifier, as represented here. True inward rectifiers are gated at the different voltages, but what is represented here are simply tail currents following a single voltage command and the only difference is in driving force. It is more of a distraction than a scientific issue since the same data could be represented at a single voltage with a bar graph, and it would be enough for this story to show differences in current at a single voltage. If a full I-V is desired, it should be created by plotting peak tail current vs. prepulse voltage.

7. The brief report on the other ion channel models was out of balance with respect to the hERG models, and was distracting. The manuscript would be improved by removing this and the associated supplemental data.

Reviewer #1 (Reviewer's comments in italics):

Jiang et al present a beautiful story that starts with the construction of worm models for four different channelopathies, performing drug screens against one of these models (hERG-LQTS2), the successful test of the worm-derived hits in human cells, and the rigorous examination of the mechanism of action of their hits in two cell-based models of the disease. I find the work compelling, exciting, of broad interest and support its publication in Nat Comm. Aside from minor errors in grammar, the manuscript itself is easy to digest and the figures are beautifully presented and easy to absorb. Despite my many comments below, I am very excited about this work/manuscript.

Major Comments:

My one major concern is focused around a secondary theme of the paper. Namely, that the hERG model models can be used to test the safety of drugs against inhibition of hERG function, which is a major concern when testing new candidate therapeutics. Because the authors found that ALA, clofilium and E-4031 can inhibit the effects of the GF version of the worm hERG model, they conclude on lines 197 and 198 that their model can be used as an inexpensive hERG safety test. There are several problems with this conclusion. First, the worm is not a human system, and despite making the worm hypersensitive to small molecule penetration, it is not possible to state with any certainty whether a molecule that shows no effects in their worm assay actually got into the worm and can reach a potential target. So, a molecule that fails to suppress the worm hERG GF model may not even be able to reach the nERG target. Second, the hERG GF model is just that, GF. So it is possible that a molecule may be able to suppress WT hERG activity, but for any number of reasons, not be able to suppress a GF type of allele. Third, its not at all clear that the worm hERG GF model phenotype has anything to do with WT ERG activity. The phenotype could be due entirely to neomorphic properties. Hence, suppression by some mechanisms will be relevant to WT ERG activity, while suppression by other mechanisms will not be relevant to WT ERG activity. IMHO, all of this is easily addressed by re-writing parts that deal with safety- either remove that part of the story entirely, or make it clear that

its proof-of-principle as a safety model, but a lot more work needs to show that its reliable.

In accordance with the reviewer's suggestion, we have removed the parts that deal with hERG safety test in both the result section and the discussion section.

Major/minor Issue: The inclusion of the four models.

First, I found the transition between the description of the various models to be jarring/abrupt. Might want to use, 'We made four different models. First, (model 1). Second, (model 2) etc' so that its clear it's a list.

Second, I didn't understand the need to include the three other models aside from hERG. They seem superfluous. Save them for other papers and tighten up this paper by removing them.

In response to the reviewer's suggestion, we have removed the data of transgenic worms expressing KCNQ2, Kv1.2, or BK channels.

Third, it would have been informative to know more about the expression pattern and biological role of the worm channels that the authors were exploiting.

In accordance to the reviewer's suggestion, we have added the information of the expression pattern and the biological role of UNC-103 potassium channel. Please refer to page 6, lines 129-132.

Fourth, if the corresponding worm gene has a loss-of-function phenotype, it would be good to know if the human (chimera or not) transgene can rescue that loss-of-function phenotype. Rescue would establish physiological relevance to the modeled human channel. Or maybe rescue doesn't matter so long as the neomorphic/GF phenotypes can be blocked with known inhibitors? Either way, the issue deserves to be addressed

in the manuscript.

Previous studies have reported that *unc-103(sy557)* males with loss-of-function mutation in *unc-103* will protract their copulatory spicules permanently within 24 hours after L4 stage¹. In accordance to the reviewer's suggestion, we have examined whether hERG^{chimera} or hERG^{chimera/A536W} channels can rescue the behavioral defects. We found that expression of hERG^{chimera} or hERG^{chimera/A536W} in *unc-103(sy557)* worms could alleviate this phenotypic defects (supplementary Fig. 1c). We have added these new data in the revised manuscript. Please refer to page 7, lines 145-147. In addition, as shown in supplementary Fig. 2b, the phenotypes in the *acs-20;hERG^{chimera/A536W}* transgenic worms were significantly ameliorated by individually adding 10 μ M well-known hERG blockers. Taken together, these data suggest that chimeric hERG channels could affect worms' behaviors.

Minor issues:

1. The writing of the results is well done. Elsewhere could use some help with grammar.

line 39- change 'identify' to 'reveals'

line 53- are the molecular basis

line 56- remains an urgent

line 71- and are known

line 143- R201C, which is found

line 331- different screening methods

line 333- 'that aiming at compounds that' needs to be revised.

line 336- 'Cumulating' needs to be revised.

line 385- "250 drugs per person per day"

We have corrected all these grammar errors. Thank you!

2. Lines 57 and 58. *That existing drugs were mostly discovered using animal models is a contentious statement and I have a hard time believing that. More literature should be investigated or this statement should be removed/modified.*

We have removed this statement. Thank you for your suggestion.

3. *It would be helpful to explain to the reader why chimeric channels were required at all. Did the unmodified human transgenes fail to elicit phenotype?*

We have tried to express full-length hERG channels in *C. elegans*, but failed due to mistrafficking of the protein. We have added this information in the revised manuscript. Please refer to page 6, lines 132-134.

4. *Line 159 etc.*

A. *It would be good to know whether the assays work in liquid (for HTP analyses) or must everything be done in solid because of the fragility of the *acs-20* mutant?*

The transgenic worms *acs-20;hERG^{chimera/A536W}* and *acs-20;hERG^{chimera/A536W/A561V}* can be cultured in the liquid system. As the egg-laying and locomotion phenotypes were easily detected on NGM plates, we performed screening assay in the solid system.

B. *Also, it would be good to explicitly state whether *acs-20* was actually needed to see suppression.*

The effects of well-known hERG blockers on *hERG^{chimera/A536W}* transgenic worms were markedly enhanced when the *acs-20* null mutation was introduced in the transgenic worms. We have added this information in the new manuscript. Please refer to pages 7-8, lines 160-164.

C. It would be good to present a rationale as to why the existing inhibitors are not sufficient therapies. loss-of-function phenotype

Although administration of hERG blockers will enhance the amount of hERG channels in the plasma membrane, the blockage action of drugs may diminish the current conducted by rescued hERG channels, consequently nullifies their therapeutic potential. We have added this explanation in the new version of manuscript. Please refer to page 16, lines 339-342.

5. Line 174. Well, the GF phenotypes indicate that they have some function, but not necessarily wt function.

In order to design a phenotype-based screening method, we introduced a gain-of-function mutation A536W in the hERG. Expression of chimeric hERG^{A536W}::GFP, but not chimeric hERG::GFP, resulted in severe behavioral defects, which offered good indicators for screening small-molecule channel modulators. Therefore, we performed phenotype-based screening for chemical regulators of the hERG channel using *acs-20;hERG^{chimera/A536W}* or *acs-20;hERG^{chimera/A536W/A561V}* transgenic worms.

Reviewer #2 (Reviewer's comments in italics):

Major Comments

The use of the c elegans model system does not provide improved clarity or efficiency in identifying new ion channel modulators and correctors. As described, the compounds that are discovered could have just as easily obtained by HTS electrophysiological approaches for HEK cells (or similar) expressing hERG channels.

Automated patch clamp systems offer a high throughput electrophysiological

approach for examining the acute effect of small molecules on the function of hERG channels expressed in HEK293 cells. Drug administration is usually conducted during the recording. The system, however, is not convenient to screen these compounds that regulating protein trafficking of ion channels because it will take at least several hours for these compounds to affect the protein trafficking. In this case, cells will be treated with drugs, cultured for hours, re-suspended, and then transferred to the automated patch clamp system for electrophysiological recording. These steps will markedly reduce the throughput of the screening. Indeed, to the best of our knowledge, so far no automated electrophysiology-based assays has been reported to screen trafficking correctors of ion channels.

In this study, we have developed phenotype-based screening assays for the discovery of small-molecule ion channel modulators using *C. elegans* channelopathy models. The screening assay provide an important tool for identifying new ion channel modulators and correctors because:

(1) Screening assays in *C. elegans* channelopathy models, where chimeric hERG channels show physiological relevance in animals, provide an in vivo system for the discovery of chemical regulators of hERG channel under physiological condition.

(2) This system provides an excellent tool for identification of functional correctors of trafficking-defective ion channels. In this study, we have successfully identified some novel trafficking correctors, illustrating the usefulness of the screening method. Cultivating *C. elegans* is easy and cheap. Behavior-based screening assays could be done manually in a regular ion channel lab with a reasonable throughput. With proper automation in behavioral screening, we believe that a high-throughput screening system based on our method will become an efficient tool for identification of correctors of trafficking-defective ion channels.

It is not clear from the data as presented if the discovered compounds impact multiple

ion channels.

In response to the reviewer's concern, we have performed additional experiments to detect the mRNA levels of hERG splice variant and other channels expressed in hiPSC^{A561V}-CMs after treatment with Prostratin or IDB. Administration of Prostratin did not affect the expression of all the channels examined, while treatment of IDB slightly enhanced the expression of *KCNQ1* and slightly reduced the expression of *Nav1.5* (supplementary Fig. 9e). We have added these data in the new manuscript. Please refer to page 14, lines 303-306.

*The possibility that site-specific phosphorylation promotes trafficking is intriguing but these data do not further the logic underlying the use of *C. elegans* for drug discovery. Overall, the approach is not well justified. In the end, the study has not advanced the understanding of channel gating or trafficking.*

Defective protein trafficking represents a common mechanism in channelopathies including LQTS, cystic fibrosis, episodic ataxia type 1, epilepsy, and hearing loss. Despite extensive interests in pharmacological correction of defective protein biogenesis of ion channels, few such compounds have been found. In this study we have made great advance in the discovery of drugs targeting the protein trafficking of ion channels:

(1) We have established an efficient in vivo screening system for the discovery of function correctors of trafficking-defective ion channels. Behavior-based screening assays in *C. elegans* models of LQTS successfully identified novel chemical modulators for hERG and its trafficking-defective mutant channels. Six novel functional correctors of LQTS mutants, including Prostratin and IDB, have been identified.

(2) We also have demonstrated that phosphorylation of S606 at the pore region could rescue the trafficking of LQTS mutant hERG channels, revealing a novel mechanism underlying protein trafficking of hERG.

Taken together, we have definitely made a significantly advance in developing new screening assays for the identification of ion channel modulators and the understanding of channel trafficking.

The logic of combining the A536W gain-of-function gating mutation with the A561V trafficking defect is unclear. The identification of Prostratin and IDB as trafficking modulators via PKC phosphorylation is new and somewhat intriguing.

In order to design a phenotype-based screening method, we introduced a gain-of-function mutation A536W in the hERG. Expression of chimeric hERG^{A536W}::GFP, but not chimeric hERG::GFP, resulted in severe behavioral defects, which offer good indicators for screening small-molecule channel modulators. To screen functional corrector of hERG mutant channels, we introduce the A561V trafficking defective mutation in hERG^{chimera/A536W} channels.

P5. 116 “Given that ion channel inhibitors and activators often target at transmembrane domains of the channels,...”. It is assumed from this statement that the authors are referring to pore-blockers? If, so then this is an over-simplification of the literature. Peptide toxins, for instance, are potent ion channel modulators that bind to loops at the membrane periphery.

We thank the reviewer for pointing out that peptide toxins bind to loops at the membrane periphery. We now have changed the sentence to "Ion channel inhibitors and activators often target at transmembrane domains and/or loops at the membrane periphery of the channels. We thus assumed that a chimeric ion channel consisting of transmembrane domains and their connecting loops of a human ion channel, and N-/C-terminus of its *C. elegans* homologue, would largely maintain pharmacological properties of the human ion channel and could be functionally expressed in *C. elegans*." Please refer to page 5, lines 112-117.

Figure 1D. Are these channels WT hERG or hERG A526W? Regardless, the utility of AHA to degrade is not clear. Simply removing some channels that have a gain-of-function gating defect would still leave a small population of mutant channels with gating defects.

I guessed this question is related to Figure 2D. In Figure 2D, we examined the effect of ALA treatment on the function of wild-type hERG channels. Sorry for the confusion, we have written it clearly in this new manuscript. Please refer to page 8, lines 175-177.

The rescued K currents in cardiomyocyte A561V cells display perturbed activation and inactivation kinetics compared to control K currents (Figure 6, panel d). These K currents seem to correct the cardiac action potential in subsequent experiments, but is it possible that in this preparation that another channel or splice variant is upregulated?

The mutation A561V impaired biophysical properties, including deactivation of hERG K⁺ channels. Administration of IDB and Prostratin partially rescued the biophysical properties of the channel (Fig. 3e and the right Figure).

In response to the reviewer's concern whether the compounds affect other channels, we have performed additional experiments to detect the mRNA levels of hERG splice variant and other channels expressed in hiPSC^{A561V}-CMs after treatment with Prostratin and IDB. Administration of Prostratin did not affect the expression of all the channels examined, while treatment of IDB only slightly enhanced the expression of *KCNQ1* and slightly reduced the expression

Figure: the effects of Prostratin and IDB on the deactivation rate (τ) of hERG^{WT}-hERG^{A561V} channels expressed in HEK293T cells.

of *Nav1.5* in hIPSC^{A561V}-CMs (Supplementary Fig. 9e). In addition, as shown in supplementary Fig. 9c and 9d, Prostratin and IDB treatment did not change the action potential duration of wild-type hIPSC-CMs, suggesting that Prostratin and IDB marginally affect the function of other ion channels. Thus, we believe that Prostratin or IDB corrects action potentials in hIPSC^{A561V}-CMs mainly because they rescued defective function of hERG^{A561V} K⁺ channels. We have added the new data in the revised manuscript.

There are major issues with grammar throughout the manuscript. For instance, first line of the Introduction "Ion channels are molecular basis for cell excitability" is missing a "the" between "are" and "molecular".

Corrected. Thank you!

Minor

The mutant in figure 1 is described as Kv1.2chimera/R297C but in the text this clone is referred to as KCNQ2chimera/R201C

In response to the other two reviewers' suggestion, we have removed this part in the revised manuscript.

Reviewer #3 (Reviewer's comments in italics):

This manuscript presents several C. elegans animal models as platforms for high-throughput screening for channel inhibitors and activators. The significance is that compounds identified in such a "native" cellular environment may be more robust

and authentic than those identified in screens performed using more reduced heterologous expression systems. For example, C. elegans screen for hERG inhibitors, it is argued, could serve as a better platform for detecting drugs in development that might cause drug-induced long QT syndrome (LQTS). A screen identifying chemical correctors of aberrant trafficking could, in principle, lead to drugs to treat the majority of inherited LQT2 cases attributed to hERG trafficking defects. The animal models were created by making chimeras between variants of C. elegans UNC-103 and hERG, its human homolog. The chimeras carry the N- and C-termini of UNC-103 and the hERG transmembrane domains, where most interacting compounds are thought to act.

Expression of the constructs were driven by the UNC-103 promoter. Overall, the model system approach is clever and could contribute new biological understanding, in addition to its utility as a screen for new drug candidates. However, there were weaknesses in the manuscript, many of which could be addressed.

1. The most intriguing findings were obtained using the heterozygous mutant A56IV, a congenital LQT2 mutation that, in heterologous systems, disrupts trafficking. Screens using this mutation identified two compounds, Prostratin and ingenol-3,20-dibenzoate (IDB), that correct the associated mutant behavior in the worms. Their effects in restoring hERG mutant currents expressed in HEK293 cells and on IKr in mutant iPSC-CMs is convincing. It would also be interesting to see the effects of these compounds on WT IKr, considering the risk associated with hERG activators in triggering short QT syndrome, an equally malignant arrhythmia.

In accordance to the reviewer's suggestion, we performed electrophysiological recordings in wild-type hiPSC-CMs to test the effects of Prostratin and IDB on WT IKr. We found that these two compounds did not affect the IKr in wild-type hiPSC-CMs. We added this data in supplemental Fig 9a and 9b.

In addition, as shown in supplemental Fig. 5 and Fig. 9, these two compounds did not affect the function of wild-type hERG K⁺ channels expressed

in HEK293T cells and the duration of action potentials of hiPSC-CMS.

2. The effect of these compounds is aptly termed "functional correction" as the behavior is restored and currents are clearly enhanced. It is difficult to know based on the evidence provided whether these compounds promote trafficking or retard degradation, since maturation measurements were not provided. It would be fairly straightforward to do this, with a measurement of the density of the fully-glycosylated 155kD band/total 135+155. This would enhance the mechanistic aspects of the manuscript.

We have measured the density of the fully-glycosylated 155 kD band/total 135 + 155 kD. We added this data in Supplementary Fig. 4. Thank you for your suggestion!

3. The great majority of LQT2 hERG mutations cause trafficking defects when expressed in HEK293 cells, but whether this is the disease phenotype in vivo has yet to be determined in most cases. In this regard it would be a contribution to include immunofluorescence data in Fig. 6 of the hERG signal. If the V561A mutant protein shows accumulation in the cytoplasm and subsequent amelioration with prostratin and IDB, this would not only add mechanistic information to the study but would provide veracity to the assumption from HEK293 cell studies that these mutations may cause trafficking aberrations even in more native-like settings.

We have performed immunofluorescence assay in control or mutant hiPSC-CMs. Compared to wild-type hERG, hERG^{A561V} proteins show accumulation in the cytoplasm. Prostratin and IDB treatment enhanced the surface expression of hERG^{A561V} and promote them trafficking to the plasma membrane. We added this data in Supplementary Fig. 8b in the revised manuscript.

4. The case for using C. elegans to screen for drugs that block hERG was not very strong. The animals must be heavily manipulated to effectively expose them to the

compound, which also seem to be in high concentration. It is hard to see this assay supplanting existing assays that are more sensitive, specific and clearly relevant to drug-induced LQTS.

In accordance to the reviewers' suggestion, we have removed the parts that deal with hERG safety test in both the result section and the discussion section.

5. The western blot in Fig. 3 showing absence of maturation by a hERG-inhibiting compound is not convincing as a sole piece of evidence. This should be presented with summary statistics for upper band maturation (as described above) over several western blots.

We have measured the density of the fully-glycosylated 155 kD band / total 135 + 155 kD, and have added the data in the Fig. 2g.

6. The way the I_{Kr} currents are represented is problematic. Although I_{Kr} has inwardly rectifying properties, it is not a classical inward rectifier, as represented here. True inward rectifiers are gated at the different voltages, but what is represented here are simply tail currents following a single voltage command and the only difference is in driving force. It is more of a distraction than a scientific issue since the same data could be represented at a single voltage with a bar graph, and it would be enough for this story to show differences in current at a single voltage. If a full I-V is desired, it should be created by plotting peak tail current vs. prepulse voltage.

We have changed the presentation of the data about I_{Kr} currents according to the reviewer's suggestion in this revised manuscript. Please refer to the Fig. 6e.

7. The brief report on the other ion channel models was out of balance with respect to the hERG models, and was distracting. The manuscript would be improved by removing this and the associated supplemental data.

We have removed the data of three other channelopathy models. Thank you for your suggestion.

1. Garcia, L.R. & Sternberg, P.W. *Caenorhabditis elegans* UNC-103 ERG-like potassium channel regulates contractile behaviors of sex muscles in males before and during mating. *J Neurosci* **23**, 2696-2705 (2003).

REVIEWERS' COMMENTS:

Reviewer #1 (Remarks to the Author):

The authors have addressed my concerns to my satisfaction. However, the manuscript could still use some help with grammar and spelling.

Reviewer #2 (Remarks to the Author):

The resubmitted manuscript is much improved however the justification of the *c. elegans* remains somewhat perplexing. In their response, the authors maintain that traditional HTS of a mammalian cell line is not feasible for the identification of small molecules which promote trafficking. The issue being, in their estimation, the long incubation times for trafficking to proceed are incompatible with electrophysiology. One counter example to this logic is the CFTR trafficking agent, lumacaftor. None the less, the manuscript will be useful and of interest to the scientific community.

Reviewer #3 (Remarks to the Author):

All my concerns have been addressed and some excellent new data have been added (unfortunately to the supplement!).

REVIEWERS' COMMENTS:

Reviewer #1 (Remarks to the Author):

The authors have addressed my concerns to my satisfaction.

However, the manuscript could still use some help with grammar and spelling.

We have corrected grammar and spelling errors in the revised manuscript.

Thanks!

Reviewer #2 (Remarks to the Author):

*The resubmitted manuscript is much improved however the justification of the *C. elegans* remains somewhat perplexing. In their response, the authors maintain that traditional HTS of a mammalian cell line is not feasible for the identification of small molecules which promote trafficking. The issue being, in their estimation, the long incubation times for trafficking to proceed are incompatible with electrophysiology. One counter example to this logic is the CFTR trafficking agent, lumacaftor. None the less, the manuscript will be useful and of interest to the scientific community.*

Thank you for support! CFTR trafficking agent lumacaftor was identified by fluorescence-based assays, but not by automated electrophysiology (PNAS. 2011 Nov 15; 108 (46): 18843-8). So far, no similar fluorescence-based assay has been established to identify hERG trafficking correctors.

Reviewer #3 (Remarks to the Author):

All my concerns have been addressed and some excellent new data have been added (unfortunately to the supplement!).

Thank for your support!